# Foaming of Bio-Based PLA/PBS/PBAT Ternary Blends with Added Nanohydroxyapatite Using Supercritical CO_2_: Effect of Operating Strategies on Cell Structure

**DOI:** 10.3390/molecules30092056

**Published:** 2025-05-05

**Authors:** Pei-Hua Chen, Chin-Wen Chen, Tzu-Hsien Chan, Hsin-Ying Lin, Ke-Ling Tuan, Chie-Shaan Su, Jung-Chin Tsai, Feng-Huei Lin

**Affiliations:** 1Department of Biomedical Engineering, National Taiwan University, Taipei 106319, Taiwan; d10528004@ntu.edu.tw; 2Department of Orthopedics, Shuang Ho Hospital, Taipei Medical University, New Taipei City 235041, Taiwan; 3Department of Molecular Science and Engineering, Institute of Organic and Polymeric Materials, National Taipei University of Technology, Taipei 106344, Taiwan; aa523383@gmail.com (T.-H.C.); barbie900608@gmail.com (H.-Y.L.); t110350011@ntut.org.tw (K.-L.T.); 4Department of Chemical Engineering and Biotechnology, National Taipei University of Technology, Taipei 106344, Taiwan; cssu@ntut.edu.tw; 5Department of Chemical Engineering, Ming Chi University of Technology, New Taipei City 243303, Taiwan; jctsai@mail.mcut.edu.tw

**Keywords:** PLA/PBS/PBAT/nHA polymer blend, supercritical CO_2_, foaming strategy, bimodal cell structure

## Abstract

This study explored the innovative foaming behavior of a novel biodegradable polymer blend consisting of polylactic acid/poly(butylene succinate)/poly(butylene adipate-co-terephthalate) (PLA/PBS/PBAT) enhanced with nanohydroxyapatite (nHA), using supercritical carbon dioxide (SCCO_2_) as an environmentally friendly physical foaming agent. The aim was to investigate the effects of various foaming strategies on the resulting cell structure, aiming for potential applications in tissue engineering. Eight foaming strategies were examined, starting with a basic saturation process at high temperature and pressure, followed by rapid decompression to ambient conditions, referred to as the (1T-1P) strategy. Intermediate temperature and pressure variations were introduced before the final decompression to evaluate the impact of operating parameters further. These strategies included intermediate-temperature cooling (2T-1P), intermediate-temperature cooling with rapid intermediate decompression (2T-2P), and intermediate-temperature cooling with gradual intermediate decompression (2T-2P, stepwise ΔP). SEM imaging revealed that the (2T-2P, stepwise ΔP) strategy produced a bimodal cell structure featuring small cells ranging from 105 to 164 μm and large cells between 476 and 889 μm. This study demonstrated that cell size was influenced by the regulation of intermediate pressure reduction and the change in intermediate temperature. The results were interpreted based on classical nucleation theory, the gas solubility principle, and the effect of polymer melt strength. Foaming results of average cell size, cell density, expansion ratio, porosity, and opening cell content are reported. The hydrophilicity of various foamed polymer blends was evaluated by measuring the water contact angle. Typical compressive stress–strain curves obtained using DMA showed a consistent trend reflecting the effect of foam stiffness.

## 1. Introduction

Tissue engineering has developed rapidly in the past four decades, with the goal of achieving the regeneration of human tissues or organs. Biomaterial scaffolds are important to multidisciplinary tissue engineering research [1,2,3]. Appropriately engineered tissues are being implanted in patients clinically [3,4,5]. Porous biomaterials are extremely useful in medical applications, where the cell structure, porosity, and interconnectivity are important for the functionality and biocompatibility of the tissue matrix. Voids in the material facilitate tissue growth and outward nutrient and waste mass transfer. Appropriate cell structures and porosities vary with tissue-specific biomaterials [6,7,8]. Porous biomaterials have a wide range of medical applications in subcutaneous tissue, bone, cardiovascular system, brain, eyes, etc., and have been reviewed in recent literature [9,10,11,12,13,14]. Biomaterials are generally divided into natural materials, metals and alloy materials, ceramic materials, polymer materials, and composite materials, each of which has its advantages and disadvantages [3,5,11,15,16]. Currently, metal-based or ceramic-based implant materials are mainly used. Polymer-based materials and their composites are more viable in the future due to their tunable, biodegradable properties and processing capabilities [17,18].

According to recent literature reports, biodegradable materials have important clinical indications, including drug delivery, tissue engineering, medical imaging, vaccine carriers, biosensing, fracture fixation, ligament reconstruction, and meniscal repair [19,20]. The synthetic biodegradable polymers most investigated are poly(ε-caprolactone) (PCL), polylactic acid (PLA), polyglycolic acid (PGA), and the copolymer of PLA and PGA [21,22]. PLA is a semi-crystalline polymer approved by the U.S. Food and Drug Administration (FDA) with desired biocompatibility and biodegradability [22]. In order to improve the shortcomings of PLA, such as toughness, hydrophilicity, and cell affinity, previous literature has studied composites of PLA with other bio-based polymers or inorganic additives. The morphology, microstructure, thermal properties, and mechanical properties of PLA with poly(butylene succinate) (PBS) or PLA with poly(butylene adipate-co-terephthalate) (PBAT) polymer blends have been studied in previous reports as potential scaffold materials in tissue engineering [23,24,25,26,27,28,29]. Inorganic bioactive materials are added to polymer blends to achieve the necessary infrastructure. For example, hydroxyapatite (HA) or nanohydroxyapatite (nHA) are major components of human bone tissue and are added to various polymer blends [30,31].

According to the literature we are aware of, most polymer scaffold materials are binary bio-based polymer blends (e.g., PLA/PBS or PLA/PBAT) or single polymers with additives such as nanohydroxyapatite (nHA) particles [24,25,26,27,28,29,30,31]. Some studies on ternary systems (e.g., PLGA/HA/collagen or PLA/PBAT/nHA) have been conducted [30,32], and more hybrid polymer/inorganic systems are expected to emerge. We recently presented thermal, physical, and mechanical studies of a novel bio-based PLA/PBS/PBAT ternary blend with added nHA [33]. Based on our previous study [33], a PLA/PBS/PBAT blend (80/15/5 by weight) with 5 wt% of added nHA met the needs of tissue engineering materials [32]. The thermal and mechanical properties of PLA/PBS/PBAT/nHA blends have been described in our previous study [33]. The added nHA helped increase the hydrophilicity and improve the blend’s strength. It was stated in our previous study [33] that the 5 wt% added nHA was uniformly distributed in the polymer blend from the TEM measurement image. This novel bio-based PLA/PBS/PBAT blend with added nHA is a promising tissue engineering material, and the foaming properties of the blend were further investigated in this study.

For tissue engineering applications, porous polymer foams accounted for the most significant market share [34]. The preferred porous material must have an open-cell morphology and interconnected cell structure. Foaming using physical blowing agents, especially carbon dioxide or nitrogen, has received much attention in academic research and industrial applications [35]. The use of supercritical carbon dioxide (SCCO_2_) for polymer foaming has been reviewed in the literature [34,35,36,37]. PLA in porous form is widely used in biomedical applications, and many studies on the use of SCCO_2_ to form porous PLA or PLA composites have been proposed in the literature [38]. Foaming polymer blends containing semi-crystalline PLA using SCCO_2_ is complex due to the CO_2_-induced crystallization of PLA [39]. Yang et al. [40] presented the foaming of PLA with SCCO_2_ at a fixed saturation pressure of 16 MPa. A foaming temperature window was proposed according to the crystal structure of PLA. Different cell size and volume expansion ratio distributions were obtained after foaming at different temperatures and pressures. Chen et al. [41] showed a similar foaming temperature window for the foaming behavior of two PLA specimens with different D-isomer contents. A two-step temperature-induced forming process was used. During the first saturation process, SCCO_2_ was saturated into PLA at 16 MPa and 150 °C for 1 h. The second foaming process was carried out at different forming temperatures, and various foaming results were analyzed to obtain the foaming temperature window. Huang et al. proposed a multiple saturation temperature method under fixed pressure [42]. They reported that in addition to the first stage of high-temperature saturation and cooling procedures, a second stage of heating and cooling was ultimately added to complete rapid decompression foaming. Various operating parameters were discussed to obtain bimodal open-cell PLA with suitable cell size for tissue engineering applications. The foaming of a polymer blend of PLA/PBS using SCCO_2_ has been presented in the literature, where the PBS phase might act as a heterogeneous nucleation site. Li et al. [43] studied the effects of PBS content and foaming temperature for PLA/PBS at a fixed foaming pressure of 10.3 MPa. The PLA/PBS blend was first heated to a higher temperature of 180 °C to eliminate thermal history and then cooled to a lower temperature for a single-stage foaming process. The temperature and material composition window to obtain open-cell results was demonstrated. A similar melt-saturated foaming strategy proposed by Wang et al. [44] was used to prepare highly expanded open-cell PLA for oil-water separation. Yu et al. studied the effect of adding PBS to PLA-based polymer blends on foaming behavior [27]. The PBS phase with low melt strength helped reduce the viscosity of the blend, and the interface between PLA and PBS acted as a cell nucleation site. It was found that PLA/PBS (80/20 by weight) could produce a highly open cell structure and obtain a bimodal cell size distribution through a two-step pressure reduction process. The foaming of PBAT was studied by Wang et al. [45] using SCCO_2_ and nitrogen co-blowing agents to improve the shrinkage of PBAT foam through a single decompression process.

The literature has also demonstrated two-step pressure-reduction foaming processes of polymers and their blends with additives [46,47,48]. The polymer sample was soaked in SCCO_2_ for a certain saturation time and then depressurized to an intermediate pressure for the first time. After being maintained at the intermediate pressure for a selected holding time, the sample was rapidly depressurized to ambient conditions to complete the foaming process. A bimodal cell size distribution could also be observed through the two-step decompression process. Large cells were obtained during the first slow pressure reduction, and small cells were received during the second rapid depressurization [47,48]. The desired large and small cell sizes for tissue engineering applications were obtained by adjusting the foaming process parameters.

To the best of our knowledge, SCCO_2_ foaming of PLA/PBS/PBAT ternary polymer blends with added nHA has not been shown in the literature. With the dispersed PBS/PBAT phase in PLA, possible nucleation sites can be expected at the interface between polymers to facilitate the foaming process. This study aims to investigate the SCCO_2_ foaming strategy of a recently reported blend of PLA/PBS/PBAT (80/15/5 by weight) with the addition of 5 wt% nHA. The main foaming parameters are the saturation and foaming temperature, pressure, and time. The foaming process included SCCO_2_ saturation into the polymer matrix, foaming at appropriate temperature and pressure, and finally, rapid decompressing to obtain the desired foamed product. The feasible operating parameters for obtaining bimodal cellular structure porous materials were studied and discussed. The innovation of this study is the foaming of a ternary polymer blend with the addition of nHA, leading to the formation of a bimodal cell structure by employing a gradual pressure reduction process.

## 2. Results and Discussion

### 2.1. Foaming Results from Strategies with (1T-1P), (2T-1P), and (2T-2P) Operating Conditions

The role of selecting appropriate operating conditions during the foaming process is significant, especially for PLA-based biodegradable polymer blends. Much previous literature has investigated the foaming temperature and pressure conditions of polymer systems containing mainly PLA. The range of operating conditions was expressed as a foaming window [39,40,41]. In a recent review article, Sarver and Kiran [35] elaborated on this forming window. As reported in these review articles, the operating temperature range was 100 to 160 °C, and the pressure range was 100 to 200 bar. The simple one-step foaming strategy involves a saturation process of SCCO_2_ absorption into polymer blends at saturation temperature (T_s_) and pressure (P_s_), followed by rapid decompression to ambient conditions, which is referred to as the (1T-1P) foaming strategy in this study.

In tissue engineering material research, there is discussion about the bimodal distribution of small and large cells [49,50], and more research on various polymer systems is still in progress. According to the literature, in order to obtain a bimodal distribution during the foaming process, a two-step operation is required by changing the temperature or pressure [27,42,46,47,51,52]. In the two-step process, variable temperature operation is the most common in the literature due to the important influence of temperature on polymer crystal formation and melt strength changes. The polymer or its blend was heated to a higher saturation temperature (T_s_) and pressure (P_s_), and maintained in the saturated state for a particular saturation time (t_s_). This step allowed SCCO_2_ to be absorbed into the polymer matrix. The system was then reduced to an intermediate or foaming temperature (T_f_) and held for a certain holding or foaming time (t_f_) before being rapidly depressurized from P_s_ to ambient conditions to complete the batch foaming process. This study refers to this as the (2T-1P) foaming strategy. If the operating pressure P_s_ dropped rapidly to an intermediate pressure level, i.e., the foaming pressure (P_f_), within the foaming time interval (t_f_) before the final rapid pressure release, we call it a (2T-2P) foaming strategy in this study.

In previous literature (PLA/PBS, the weight ratio of 80/20) [27], the saturation temperature (T_s_) used was 150 °C. It is also noticed that at higher T_s_ above 160 °C, the melt strength of PLA was insufficient to withstand the internal pressure during the absorption process [41]. Therefore, the saturation temperature T_s_ in this study was selected as 150 °C. Based on the solubility data of SCCO_2_ in biodegradable polymers measured in previous literature [53,54,55], the saturation time (t_s_) was selected as 90 min, at which the CO_2_ uptake in the PLA-based blends in this study was estimated to be close to equilibrium. This saturation time is reasonable compared to the values used in previous literature [40,41,56,57,58]. Firstly, the (1T-1P) foaming strategy was adopted, and the experiments were carried out at a saturation temperature of 150 °C and saturation pressures of 130, 150, and 170 bar. Based on these preliminary results, the highest expansion ratio can be obtained when the saturation pressure (P_s_) was selected as 150 bar, and it is reasonable to set P_s_ as 150 bar in this study.

Figure 1 shows the operating conditions for three foaming strategies and their resulting SEM images: (A) one-step (1T-1P) process, (B) two temperature levels (2T-1P) process with intermediate cooling to 110 °C, holding time was 10 min, and (C) two temperature levels and two pressure levels (2T-2P) process, with intermediate temperature cooling to 110 °C and intermediate rapid pressure reduction from 150 to 100 bar. The intermediate pressure at 100 bar was selected based on the foaming window shown in the literature [35]. The intermediate temperature at 110 °C chosen in this study is reasonable, which is about the middle temperature in the previously reported foaming window [40,41]. A foaming time t_f_ of 10 min was used in the (2T-1P) and (2T-2P) strategies.

From the SEM observation of the (1T-1P) foaming strategy A, it is evident that the foamed product contained sparsely distributed small cells, while a significant portion of the sample remained unfoamed. This outcome suggests that during the (1T-1P) foaming process, only a small amount of absorbed CO_2_ underwent nucleation to form small bubbles during the rapid pressure drop, leaving behind a large unfoamed region. The defining feature of the (1T-1P) strategy is its isothermal operation, meaning that the polymer blend absorbed CO_2_ at a single elevated temperature and then initiated foaming by a sudden release of pressure to ambient conditions. However, due to the lack of a second temperature zone to adjust the melt strength and foaming ability, the PLA/PBS/PBAT/nHA blend could not form a large or bimodal cell structure. This limitation is particularly critical for polymer blends, where achieving a well-defined porous structure depends on carefully controlling both temperature and pressure conditions. Without an additional temperature change stage, the melt strength remained inappropriate to support stable bubble growth, leading to the unsatisfactory formation of only small cells with an average size of 27.6 μm using strategy A.

The (2T-1P) foaming strategy B shown in Figure 1 included an additional temperature cooling (from 150 to 110 °C) and holding at 110 °C for 10 min. During the cooling and holding process, bubbles nucleated and presented an unimodal cell structure after rapid decompression. With foaming strategy B, no large cells were detected from the SEM images. This may be because the nucleation energy barrier was not overcome within the holding time to induce the nucleation and cell growth. Only during the final rapid decompression step did relatively short nucleation and small-cell formation occur [59,60]. Small cells with an average size of 34.2 μm were obtained through foaming strategy B. To obtain both small and large cells for tissue engineering applications, consideration of two-step pressure manipulation was further investigated.

The (2T-2P) foaming strategy C shown in Figure 1 included temperature cooling (from 150 to 110 °C) and holding at 110 °C for 10 min. As the temperature reached 110 °C, the pressure was quickly reduced from 150 to 100 bar and held in this intermediate pressure for 10 min before final rapid depressurization to ambient conditions. This foaming strategy produced mainly large cells with an average size of 408.7 μm and a lower cell density. The cell size and structure produced by strategy C are completely different from those by strategy B, mainly due to a rapid, intermediate pressure drop from 150 to 100 bar. According to classical nucleation theory [48,61], the free energy barrier for nucleation is inversely proportional to the square of pressure supersaturation, which is defined as the difference between the critical bubble pressure and the operating pressure. When the operating pressure was quickly reduced by 50 bar, it is possible that excessive cell nucleation was induced during the quick, intermediate decompression and a high degree of cell growth during the holding time, resulting in almost no small cell size during the final rapid decompression to ambient conditions. SEM images showed large cells, indicating excessive expansion of the bubbles and possible coalescence. This result expresses that larger cells can be obtained through a two-step depressurization operation, but the degree of the intermediate depressurization and the time of holding the intermediate pressure should be adjusted to obtain a bimodal cell structure.

Figure 2 plots the relative frequency of cell size distribution for three foaming strategies A, B, and C. It shows that unimodal cell structures were obtained from these three strategies. This study observed that the (1T-1P) foaming strategy A had a lower expansion ratio of 1.36 and a smaller cell size. The (2T-1P) foaming strategy B increased the expansion ratio to 2.72, but the cell size was still small. The (2T-2P) foaming strategy C, which changed both temperature and pressure in the intermediate stage, produced almost all large cells with a higher expansion ratio of 5.41. The intermediate pressure change is beneficial for obtaining larger cells, but a gradual pressure change might be needed to get desirable bimodal cell structure products.

### 2.2. Foaming Results from Strategies with (2T-2P, Stepwise ΔP) Operating Conditions: Pressure Effect

Based on the three foaming strategies discussed above, the temperature change process accompanied by a stepwise decrease in pressure (2T-2P, stepwise ΔP) was further investigated. Figure 3 shows a schematic diagram of the (2T-2P, stepwise ΔP) foaming strategy. The saturation pressure and temperature remained at 150 bar and 150 °C, respectively. The saturation time was maintained at 90 min to ensure sufficient absorption of CO_2_ into the polymer blends. After the saturation stage, the operating temperature drops to an intermediate value (T_f_) (or the foaming temperature) and remains there for a holding time (t_f_). During this holding time, the operating pressure was gradually reduced from 150 bar to an intermediate value (P_f_) (or the foaming pressure), with each pressure reduction step being 10 bar min^−1^. With the gradual change in pressure, a porous product with a bimodal cell structure is expected to be obtained, in which large cells are formed during the intermediate holding time, and small cells are formed during the final rapid decompression step [27,46,47,48].

Firstly, the effect of foaming pressure (P_f_) was studied, where the foaming temperature (T_f_) was maintained at 110 °C. Three (2T-2P, stepwise ΔP) foaming strategies were investigated based on different intermediate pressure effects: (D) stepwise pressure drop to 120 bar with a holding time (t_f_) of 3 min, (E) stepwise pressure drop to 100 bar with a holding time of 5 min, and (F) stepwise pressure drop to 80 bar with a holding time of 7 min. Figure 4 shows the SEM images and relative frequencies of cell size distribution obtained for the three foaming strategies: D, E, and F.

It is observed that bimodal cell structures were obtained with the (2T-2P, stepwise ΔP) foaming processes. Generally speaking, the average small cell size ranged from 110 to 160 μm for the three foaming strategies: D, E, and F. However, the average large cell size depended on the degree of pressure change. Both the operating temperature and pressure have an important influence on the cell size and structure. For foaming strategies D, E, and F, the intermediate temperature (T_f_) was maintained constant at 110 °C. The effects of the remaining operating parameters, namely the degree of pressure variation and the foaming time (t_f_), are discussed below.

For these three foaming strategies at the same intermediate temperature, as the pressure decreased in the intermediate foaming stage, the solubility of SCCO_2_ in the polymer becomes smaller, resulting in the gradual release of gas from the polymer matrix. When CO_2_ escaped, it created a supersaturated state, which caused bubbles to form. The extent of the pressure drop affected the number and size of bubbles. The pressure drop extent of foaming strategy D was the smallest among the three pressure reduction strategies, and the resulting average large cell size was also the smallest, about 562 μm.

This result is consistent with the classical nucleation theory, where the maximum free energy barrier related to forming a new phase is inversely proportional to the square of pressure supersaturation. Cell nucleation and growth occurred during the time when the pressure was gradually reduced. The pressure drop of 30 bar in foaming strategy D lasted for 3 min, which was insufficient to allow the growing nuclei to form larger cell sizes compared to the other two foaming strategies, E and F, with longer foaming times. If a larger cell size is desired, the degree of pressure drop should be increased.

The results of foaming strategy E show that when the intermediate pressure was reduced to 100 bar and the foaming time was extended to 5 min, larger cells could be obtained, and the average cell size was about 602 μm. The results for strategy F showed a gradual pressure drop to 80 bar over a period of 7 min, with an average large cell size of 775 μm. A pressure drop of 70 bar in foaming strategy F might reduce the nucleation barrier to a low enough level to result in the growth of large cells. Examining the pressure effects shown in Figure 4, it can be noted that intermediate stepwise pressure drops can adjust the desired cell size and bimodal cell structure. Various cell sizes might be required for different applications in tissue engineering [6]. For example, bimodal cell structure is advantageous for bone tissue engineering scaffolds, and large cell sizes up to 800 μm still contribute to bone formation [62]. Figure 5 plots the average large cell sizes for foaming strategies D, E, and F. It supplies a guideline for choosing the operating pressure condition to obtain desired bimodal cell structures.

### 2.3. Foaming Results from Strategies with (2T-2P, Stepwise ΔP) Operating Conditions: Temperature Effect

The effect of foaming temperature (T_f_) was further examined, with the foaming pressure (P_f_) gradually reduced from 150 to 100 bar, as described in Section 2.2 under foaming strategy E. Three variations of the (2T-2P, stepwise ΔP) foaming strategy were investigated, each with a different intermediate or foaming temperature (T_f_): (G) T_f_ set at 100 °C, (E) T_f_ set at 110 °C, and (H) T_f_ set at 120 °C. The foaming time (t_f_) was maintained at 5 min for all three strategies. Figure 6 presents the SEM images and corresponding cell size distribution frequencies for strategies G, E, and H. Under the same foaming pressure conditions, temperature emerged as a critical factor influencing the solubility and diffusivity of supercritical CO_2_ within the polymer blend. It also affected the molecular chain mobility and melt strength during the foaming process [52,63], ultimately impacting the resulting cell structure.

It was observed that all three foaming strategies produced a bimodal cell structure, with average small cells ranging from 105 to 164 μm. However, the average large cell sizes varied under each foaming temperature condition. All three temperatures fell within the foaming window of PLA, as reported in previous studies [39,40,41], making them suitable for foaming the PLA-based blends in this study. For strategies G, E, and H, the results show a decreasing trend in macrocell size as the foaming temperature changed from 100 °C to 120 °C. This trend can be attributed to the changes in SCCO_2_ solubility within the PLA-based blends of this study and the effects on the melt strength of these blends.

Foaming strategy G, operated at a foaming temperature (T_f_) of 100 °C, produced the largest average macrocell size of 889 μm among the three strategies. This result can be explained first by gas solubility. In strategy G, the larger temperature drop from the high saturation temperature to the lower foaming temperature resulted in a higher gas density, increasing the solubility of SCCO_2_ in the polymer blend. During stepwise decompression, higher SCCO_2_ contents led to greater supersaturation, generally promoting larger cell formation. In addition to SCCO_2_ solubility, the melt strength of the polymer blend played an important role. According to the literature [52], sufficient melt viscosity can stabilize cell growth. At 100 °C, the polymer blend exhibited relatively higher melt strength than those at the other two foaming temperatures, which benefited the bubble expansion and, thus, the formation of larger macrocells. This explains the result observed with strategy G, where the combination of melt strength effect and stepwise pressure reduction created a favorable environment for large cell growth, ultimately producing a bimodal cell structure after the final decompression.

Foaming strategies E and H were conducted at higher foaming temperatures (T_f_) of 110 °C and 120 °C, respectively. This resulted in a smaller temperature drop from the initial saturation temperature compared to strategy G. The smaller temperature drop maintained the SCCO_2_ density lower than that of strategy G. The lower solubility of SCCO_2_ may have resulted in the reduction in the macrocell size compared to the 100 °C condition. Additionally, the melt strength of the polymer blend decreased at these higher foaming temperatures. During the final rapid cooling and sudden drop in pressure to ambient conditions, the melt strength of the polymer blend increased again. As reported in the literature [40,64], the sudden increase in melt strength might promote the retraction of polymer chains, leading to bubble shrinkage and further reducing the final macrocell size. As shown in Figure 6, the average macrocell sizes for foaming strategies E and H were 602 μm and 476 μm, respectively. A graphical illustration of the decreasing trend of the average macrocell size from strategies G, E, and H is shown in Figure 7.

### 2.4. Comparison of Foaming Results from All Operating Strategies in This Study

The experimental results of all foaming strategies in this study are summarized in Table 1. It was observed that foaming strategies A and B produced only small cells with relatively low expansion ratios (less than 3) and high cell densities. The opening cell contents of foaming strategies A and B are lower because the operating pressure drops too quickly, resulting in larger unfoamed spaces and more closed cells. When applying foaming strategy C involving a rapid, intermediate pressure drop of 50 bar, only large cells with an increased expansion ratio of up to 5.41 were obtained. The opening cell content increased to 67.8, possibly due to cell coalescence when the cells grew to a large size.

The remaining five foaming strategies all resulted in bimodal cell distribution, which is advantageous in tissue engineering and regenerative medicine applications. Comparing foaming strategies D, E, and F, the expansion ratios increased from 4.3 to 14.0 as the intermediate pressure drop increased. The macrocellular size (from 562 ± 29.4 to 775 ± 13.0 μm) and opening cell content (from 60.2 ± 1.2 to 77.8 ± 0.6%) also showed an increasing trend. Larger cells indicate that the bubbles may have merged during the expansion process, forming interconnected cells and leading to higher opening cell content. The macrocell sizes obtained from strategies G and H showed their dependence on the intermediate foaming temperature. In this study, the largest macrocell size of the foamed product prepared by strategy G at the intermediate temperature of 100 °C was 889.8 ± 66.2 μm, and the highest opening cell content was 84.3 ± 0.4%. Foaming strategy H, which was operated at a higher intermediate foaming temperature of 120 °C, had a smaller macrocell size of 476.4 ± 23.6 μm and a smaller opening cell content of 57.8 ± 1.2% compared to strategy G.

The graphical presentations of the large and small cell sizes, cell densities, expansion ratios, and porosities, and opening cell contents of the foamed polymer blends in this study using various foaming strategies are illustrated in Figure 8a–d, respectively. It is demonstrated that with the stepwise pressure drop in the intermediate stage of the foaming process, bimodal cell structures can be achieved with porosity ranging from 76.7% to 94.4% and opening cell content from 57.8% to 84.3%. These data suggest that this study’s foamed polymer blends are suitable for bioscaffold use in tissue engineering [5,6,13].

The water contact angle measurements demonstrate the hydrophilicity of the foamed polymer blends produced using various foaming strategies. Figure 9 presents a representative water contact angle result for the blend processed with foaming strategy E. The initial water contact angle was recorded as 68.5 ± 0.2°, gradually decreasing to 50.7 ± 0.1° after 30 min. In comparison, the contact angle of the unfoamed PLA/PBS/PBAT/nHA blend was 87.5 ± 1.0°, which was close to the hydrophobic nature. These results indicate that the bimodal cell structure achieved through foaming strategy E expressed enhanced hydrophilicity. Similarly, all other foamed products with a bimodal structure displayed a comparable trend in hydrophilicity, attributed to the formation of large open cells that facilitated water absorption.

The decreased behavior of the water contact angle with time is discussed below for the foamed polymer blends treated using the (2T-2P, stepwise ΔP) foaming strategies described in Section 2.2 and Section 2.3. All of these foams had a bimodal cell structure, and the results show that the foamed products with larger macrocells had lower water contact angles. For example, as shown in Figure 10a, the initial water contact angles of the foamed products from strategies D, E, and F were 76.0°, 68.5°, and 54.4°, respectively. This is because strategy F produced the foamed products with larger macrocells and a larger opening cell content than the foamed products produced by strategies D and E. After a period of 30 min, the water contact angle of the foamed product produced by strategy F decreased to 27.4°, which was lower than the water contact angles of the foamed products produced by strategies D and E (61.4° and 50.7°, respectively). Figure 10b shows similar trends for the foamed products prepared using strategies G, E, and H. The decreasing water contact angle curves shown in Figure 10a,b highlight the enhanced hydrophilicity of the bimodal foamed products due to interconnected macrocells.

Figure 11 shows the DMA measurement results of the foamed products processed using the (2T-2P, stepwise ΔP) foaming strategy described in Section 2.2 and Section 2.3. All foamed products had a bimodal structure. Figure 11a presents the compressive stress–strain curves of the foamed products produced using foaming strategies D, E, and F. The strain data were recorded using dynamic mechanical analyzer (DMA, 7e, Perkin Elmer, Waltham, MA, USA) at a constant increasing load stress rate of 10 kPa min^−1^, starting from 7 kPa up to 1.13 MPa. The foamed product from strategy F displayed a greater strain value under fixed compressive stress than those from strategies D and E. This is consistent with the increasing trend of average macrocell size obtained from these strategies. Specifically, the foamed product from strategy D had a compressive stress of 0.76 MPa at 5% strain. For the foamed products from strategies E and F, at the same 5% strain value, the corresponding compressive stress dropped to 0.45 MPa and 0.12 MPa, respectively. These results indicate that the foamed product with smaller macrocells using strategy D exhibited higher stiffness, whereas the foamed products with increasing macrocell sizes using strategies E to F exhibited better toughness. Figure 11b shows a similar trend for the foamed products produced using strategies G, E, and H, indicating that the foamed product using foaming strategy H (with macrocell size of 476.4 ± 23.6 μm) exhibited higher stiffness. In contrast, the foamed product prepared using foaming strategy G (with a macrocell size of 889.8 ± 66.2 μm) exhibited enhanced toughness.

## 3. Materials and Methods

### 3.1. Material

The materials used in this study are the same as those in our previous research [33], and according to the supplier, the main physical properties of these materials are as follows. Polylactic acid (PLA, Ingeo 4032D), with an average d-lactide content of PLA of 1.4 wt%, was purchased from NaturalWorks LLC, Minnetonka, MN, USA. The melting temperature of PLA is 155–170 °C, and its melt flow index (at 190 °C and 2.16 kg) is 7 g/10 min. Poly(butylene succinate) (Bio PBS, FZ 91) was purchased from PTT MCC Biochem Co. Ltd., Bangkok, Thailand. The melting temperature of PBS is 115 °C, and its melt flow index (at 190 °C and 2.16 kg) is 5 g/10 min. Poly(butylene adipate-co-terephthalate) (PBAT, ecoflex F blend C1200) was purchased from BASF SE, Ludwigshafen, Deutschland. The melting temperature of PBAT is 110–120 °C, and its melt flow index (at 190 °C and 2.16 kg) is 2.7–4.9 g/10 min. Nano-hydroxyapatite (nHA, CAS registry number 12167-74-7) was purchased from Sigma-Aldrich, UNI-ONWARD Corp, Taiwan. The purity of nHA is greater than 97 wt%, with a mean particle size of 72–80 nm, molecular weight of 502.3, and a melting temperature of about 1100 °C.

### 3.2. Preparation of Composite Blends

The polymer blends (PLA/PBS/PBAT) added with nHA were prepared using a twin-screw extruder (Process 11, Thermo Fisher Scientific, Waltham, MA, USA). All PLA, PBS, and PBAT polymers were first dried in a vacuum oven (Channel, VO45L, KO TSAO specialty Instrument & Supplies Co. Ltd., Taipei, Taiwan) at 80 °C for 6 h before charging into the extruder. The formulation of the blends PLA/PBS/PBAT used in this study is 80/15/5 in weight percentage, added with an additional 5 wt% nHA. The extruder for blending the polymers was equipped with a volumetric feeder and a strand pelletizer (Process 11, Thermo Fisher Scientific, Waltham, MA, USA). The diameter of the screw extruder is 11 mm, and the L/D ratio is 40. Polymers were fed into the hopper of the extruder, where the extrusion temperatures were controlled independently. The feed rate of the polymer pellets was 1 kg h^−1^, and the screw speed was 50 rpm. The temperature settings for the feed and mixing zones were 220–225–225–225–225–220–210–200 °C. After the blending process, the extruded products were cooled by a water bath and were then granulated and dried for a sufficient time. Rectangular specimens with 3 mm × 7 mm and thickness of 2 mm were made by compression molding at 200 °C for 5 min, followed by cooling to room temperature. The specimens were packed in plastic bags and stored in cool surroundings before the foaming experiments.

### 3.3. Saturation and Foaming Steps

Saturation and foaming steps were included to obtain a porous structure of the polymer blend. Supercritical CO_2_ diffused into the polymer blend at high pressure (saturation pressure, P_s_) and high temperature (saturation temperature, T_s_) with a saturation time of t_s_. The foaming stage involved changes in intermediate temperature and pressure, namely, foaming temperature (T_f_), foaming pressure (P_f_), and foaming time (t_f_), and finally, rapid decompression to ambient conditions to obtain the final foamed product. A schematic diagram for the foaming process of the PLA/PBS/PBAT/nHA samples in this study is shown in Figure 12. The polymer blend sample was put into a high-pressure vessel (Applied Separations 70770, Allentown, PA, USA). The chamber was placed inside a self-made electric heater with an openable cover and temperature controller. The uncertainty of temperature was ±2 °C in this study. The chamber was purged with low-pressure CO_2_ for 5 min to remove any air inside. The chamber was heated to the desired temperature by the electric heater and pressurized to the operating pressure using an ISCO 260D syringe pump (Teledyne Technologies, Lincoln, NE, USA). This study emphasized investigating various strategies of saturation and foaming of the PLA/PBS/PBAT polymer blends added with nHA. The detailed temperature and pressure operating parameters for various foaming strategies are listed in the Section 2.

### 3.4. Determination of the Characteristic Structures and Properties of Foamed Polymer Blends

The characteristic structures of the fracture surfaces of the foamed polymer blends were observed using a scanning electron microscope (SEM, Nova NanoSEM 230, Hillsboro, OR, USA). Prior to SEM measurements, the foamed samples were dried under a vacuum and sputtered with gold.

The expansion ratio φ of the foamed sample was determined by Equation (1) [27,41]:(1)$$\varphi =\frac{\rho_{s}}{\rho_{f}}$$
where $\rho_{s}$ and $\rho_{f}$ are the densities of the solid (unfoamed) and foamed samples, respectively. These densities were measured by the water replacement method according to ASTM D792 [55,57].

The cell density *N*_0_ (cells/cm^3^) is the number of cells per unit volume (cm^3^) of foamed polymer, was determined from Equation (2) [27,40,41]:(2)N0=6[1−ρfρs]πD3×1012
where *D* is the average cell diameter (μm) measured from the SEM images.

The porosity of the foamed samples was evaluated using the expansion result by Equation (3) [27,41]:(3)$$\varepsilon =\frac{\varphi -1}{\varphi }$$

The opening cell content (OCC) is an important factor for the tissue material. It is determined by the ratio of the open-cell volume ($V_{open})$ to the total volume ${(V}_{total})$ of the foamed sample, as shown by Equation (4) [43]:(4)$$OCC=\frac{V_{open}}{V_{total}}=1-\frac{V_{true}}{V_{total}}$$

The open-cell volume is calculated by subtracting the true volume ${(V}_{true}$, the close-cell volume plus the cell wall volume) from the total volume. The open-cell volume values were measured by a nitrogen pycnometer (AccuPyc II 1340, Micrometric, Norcross, GA, USA).

### 3.5. Water Contact Angle and Mechanical Compression Measurements

Water contact angle measurements for various foamed polymer blends were performed using a contact angle analyzer (SEO Phoenix, S.E.O. Co. Ltd., Ansung City, Republic of Korea). The initial water contact angle and its changes with time were recorded for up to 30 min. The compressive stress–strain data of the foamed polymer blends were obtained using a dynamic mechanical analyzer (DMA, Perkin Elmer 7e, Waltham, MA, USA). The specimens were tested under a loading range from 7 kPa to 1.13 MPa at a constant rate of 10 kPa min^−1^. A similar DMA experiment for porous PDLLA/bioglass composites has been shown in the literature [65].

## 4. Conclusions

In this study, supercritical CO_2_ was used as an environmentally friendly physical foaming agent to investigate the foaming behavior of biodegradable PLA/PBS/PBAT ternary blends with the addition of nanohydroxyapatite (nHA). Eight different foaming strategies were explored, varying operating temperature, pressure, and duration. It was observed that intermediate temperature adjustments and pressure drop regulations are critical before the final rapid decompression to ambient conditions, ensuring a foamed product with an acceptable expansion ratio and cell size.

A stepwise intermediate decompression strategy was employed to produce foams with a bimodal cell structure. Under different foaming temperatures and pressures, the average size of the smaller cells in the bimodal structure ranged from 105 to 160 µm, while the large cells ranged from 476 to 889 µm. The gas solubility principle and classical nucleation theory were applied to explain the effects of stepwise pressure reduction. This study found that with the increase of the intermediate pressure drop, the nucleation energy barrier decreased, resulting in the bimodal structure’s enlargement of the macrocell size. Furthermore, increasing the intermediate foaming temperature led to a decrease in the size of the larger cells. The temperature effect was attributed to the lower CO_2_ solubility at higher intermediate temperatures and the possibly more pronounced polymer retraction effect during the final rapid decompression stage. The stepwise pressure variation strategy produced foams with diverse cell sizes and opening cell contents ranging from 60% to 84%, making them suitable for tissue engineering applications.

Water contact angles were measured for each foamed polymer blend, and their changes over 30 min were recorded. The results confirmed that all foamed products exhibited hydrophilic behavior, and the hydrophilicity increased with the increase in macrocell size and opening cell content. Compressive stress–strain measurements were conducted using a dynamic mechanical analyzer (DMA). Mechanical results show that the foamed polymer blends with smaller macrocells had higher stiffness, while those with larger macrocells showed higher toughness.

## Figures and Tables

**Figure 1 molecules-30-02056-f001:**
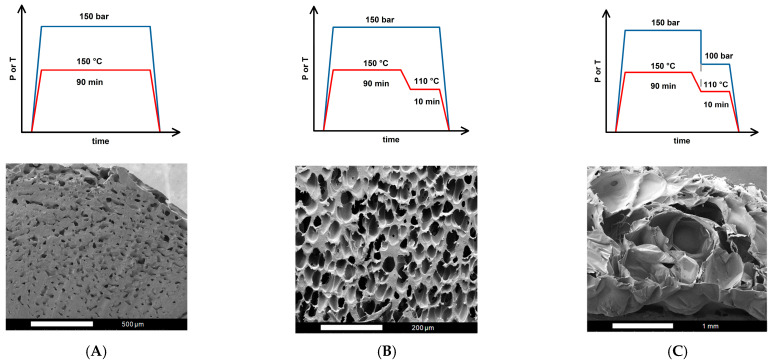
Operating conditions and SEM images for foaming strategies: (**A**) Strategy A with one-step (1T-1P) process, (**B**) Strategy B with two temperature levels (2T-1P) process, with intermediate cooling to 110 °C, and (**C**) Strategy C with two temperature levels and two pressure levels (2T-2P) process, with intermediate cooling to 110 °C and intermediate rapid pressure reduction to 100 bar.

**Figure 2 molecules-30-02056-f002:**
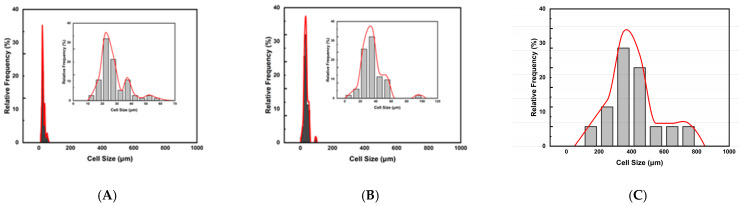
Plots of the relative frequencies of cell size distribution results for foaming strategies: (**A**) Strategy A with one-step (1T-1P) process, (**B**) Strategy B with two temperature levels (2T-1P) process, with intermediate cooling to 110 °C, and (**C**) Strategy C with two temperature levels and two pressure levels (2T-2P) process, with intermediate cooling to 110 °C and intermediate rapid pressure reduction to 100 bar.

**Figure 3 molecules-30-02056-f003:**
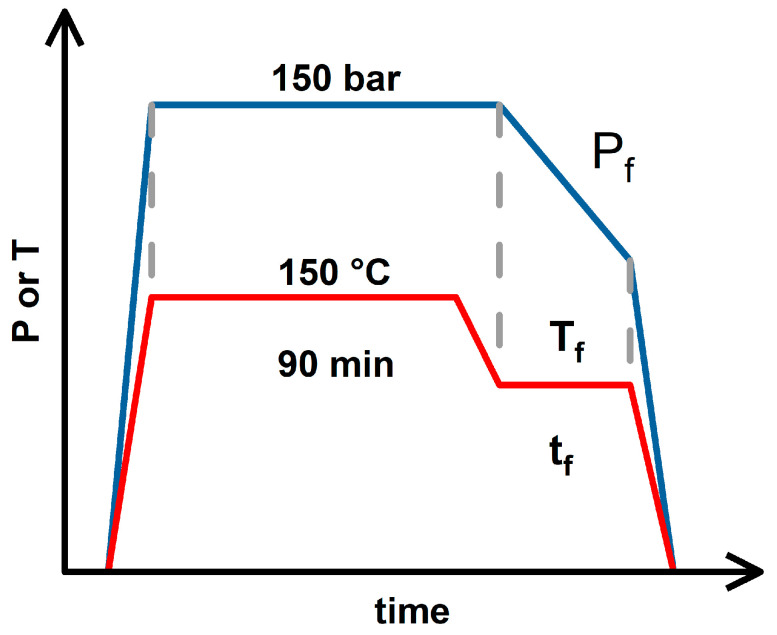
The schematic diagram for operating conditions of the (2T-2P, stepwise ΔP) foaming strategy.

**Figure 4 molecules-30-02056-f004:**
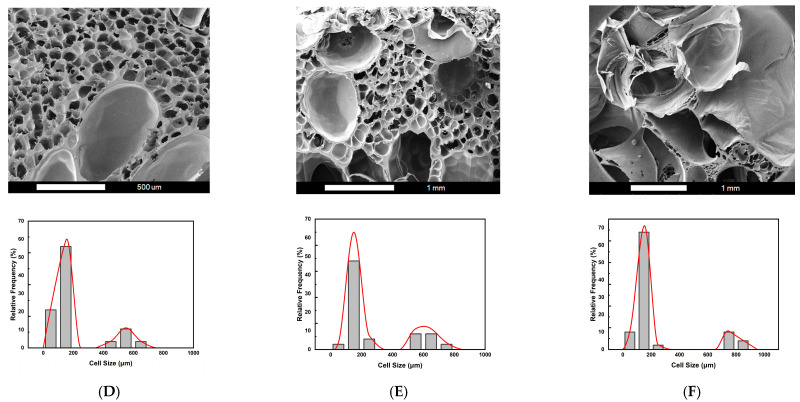
The SEM images and relative frequencies of cell size distribution results for (2T-2P, stepwise ΔP) foaming strategies: (**D**) Strategy D with stepwise pressure drop to 120 bar and with a holding time of 3 min, (**E**) Strategy E with stepwise pressure drop to 100 bar and with a holding time of 5 min, and (**F**) Strategy F with stepwise pressure drop to 80 bar and with a holding time of 7 min.

**Figure 5 molecules-30-02056-f005:**
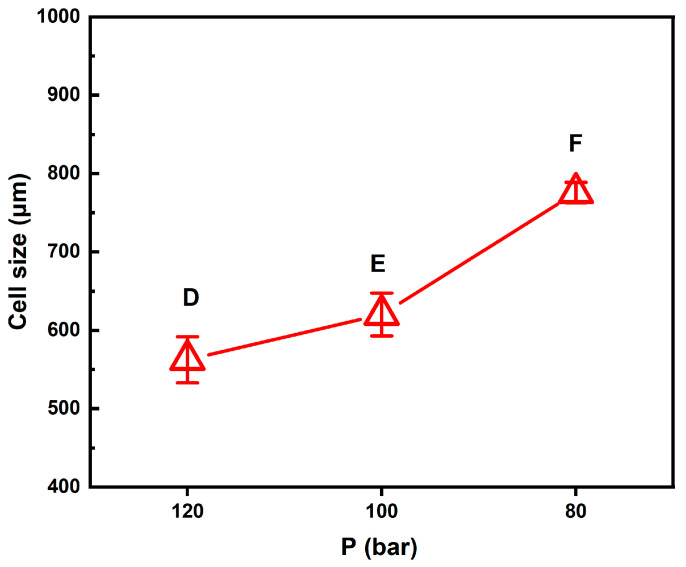
Effect of the foaming pressure on the large cell size obtained from foaming strategies D, E, and F.

**Figure 6 molecules-30-02056-f006:**
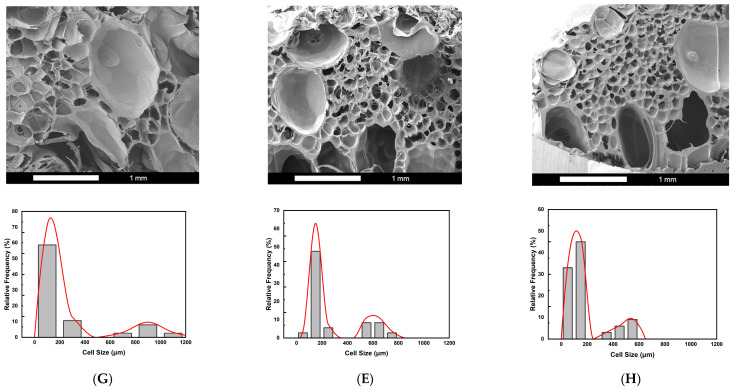
The SEM images and relative frequencies of cell size distribution results for (2T-2P, stepwise ΔP) foaming strategies: (**G**) Strategy G with intermediate temperature at 100 °C, (**E**) Strategy E with intermediate temperature at 110 °C, and (**H**) Strategy H with intermediate temperature at 120 °C.

**Figure 7 molecules-30-02056-f007:**
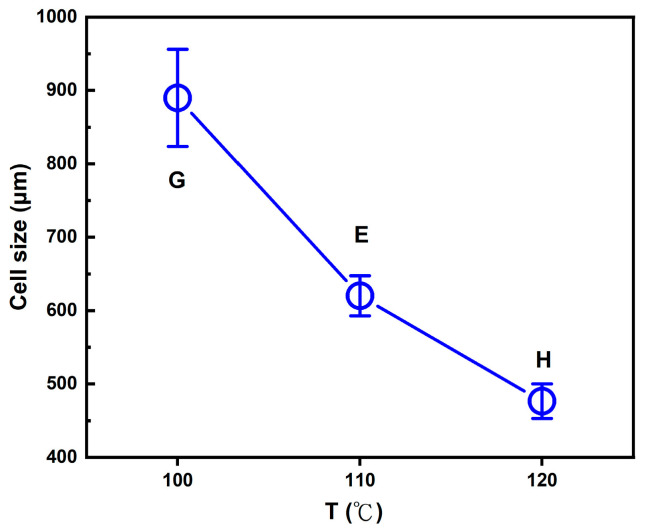
Effect of the foaming temperature on the large cell size obtained from (2T-2P, stepwise ΔP) foaming strategies G, E, and H.

**Figure 8 molecules-30-02056-f008:**
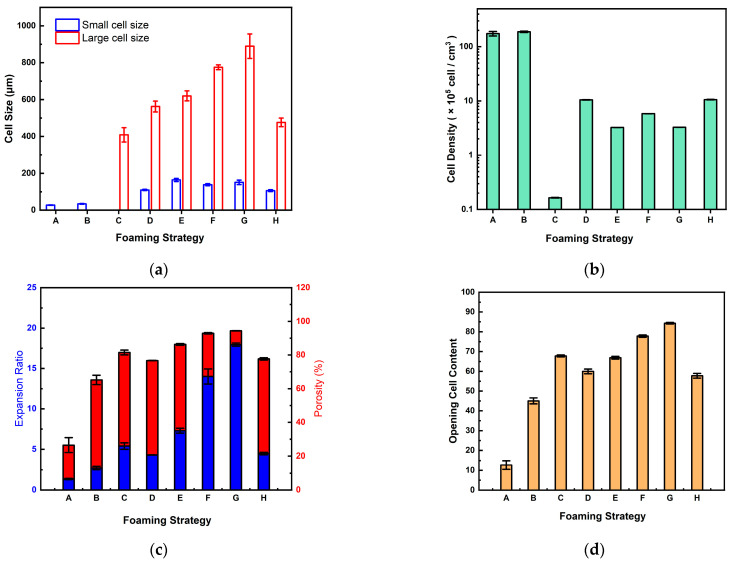
The experimental results of (**a**) cell size, (**b**) cell density, (**c**) expansion ratio and porosity, and (**d**) opening ratio of the foamed polymer blend in this study using various foaming strategies from A to H.

**Figure 9 molecules-30-02056-f009:**
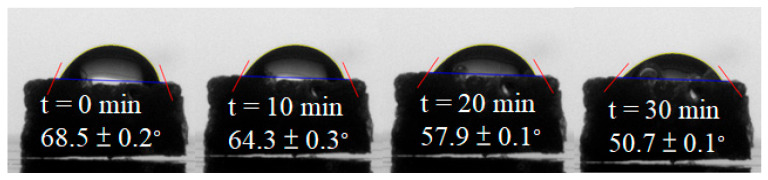
The water contact angle of the foamed polymer blend using foaming strategy E.

**Figure 10 molecules-30-02056-f010:**
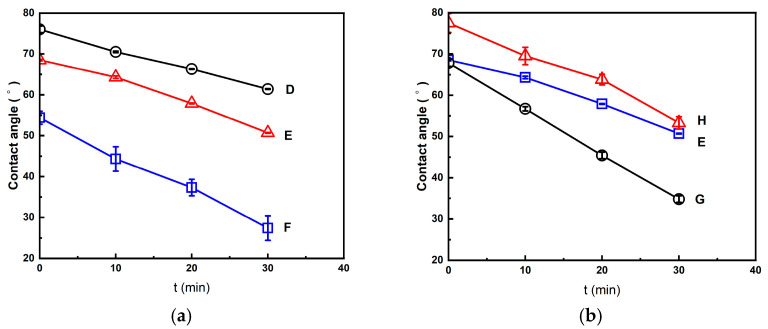
The decreasing trend of water contact angle for the foamed polymer blends from (**a**) foaming strategies D, E, and F; (**b**) foaming strategies G, E, and H.

**Figure 11 molecules-30-02056-f011:**
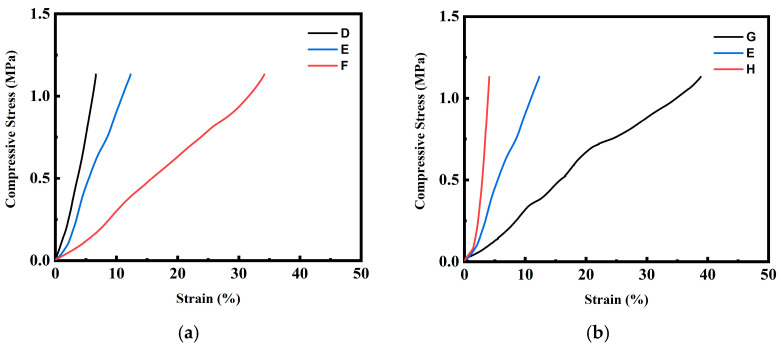
The compressive stress–strain curves for the foamed polymer blends from (**a**) foaming strategies D, E, and F; (**b**) foaming strategies G, E, and H.

**Figure 12 molecules-30-02056-f012:**
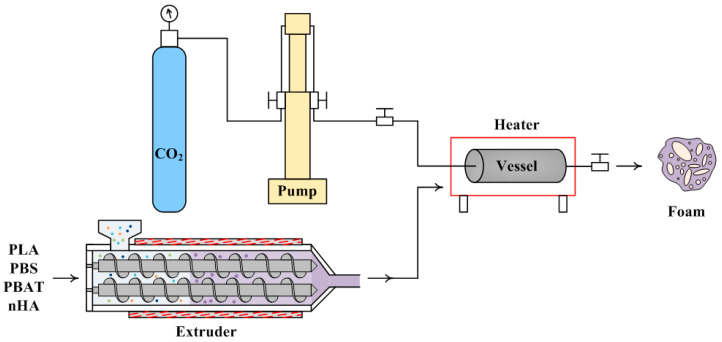
Schematic diagram of the experimental foaming process.

**Table 1 molecules-30-02056-t001:** Summarized foaming results from all operating strategies in this study.

Operating Strategy	Expansion Ratio	Porosity (%)	Cell Density (cell/cm^3^)	Cell Size (μm) (Small/Large)	Opening Cell Content (%)
A	1.36 ± 0.08	26.5 ± 4.4	(1.75 ± 0.17) × 10^7^	27.6 ± 1.1/―	12.7 ± 2.2
B	2.72 ± 0.21	65.2 ± 2.8	(1.89 ± 0.05) × 10^7^	34.2 ± 1.9/―	45.0 ± 1.5
C	5.41 ± 0.40	81.5 ± 1.4	(1.64 ± 0.02) × 10^4^	―/408.7 ± 38.7	67.8 ± 0.5
D	4.30 ± 0.02	76.7 ± 0.1	(1.05 ± 0.01) × 10^6^	109.9 ± 3.2/562.0 ± 29.4	60.0 ± 1.2
E	7.29 ± 0.30	86.3 ± 0.5	(3.24 ± 0.01) × 10^5^	164.2 ± 8.4/602.1 ± 27.3	66.9 ± 0.7
F	14.01 ± 0.93	92.9 ± 0.4	(5.86 ± 0.02) × 10^5^	138.2 ± 5.4/775.6 ± 13.0	77.8 ± 0.6
G	17.96 ± 0.20	94.4 ± 0.1	(3.27 ± 0.01) × 10^5^	151.1 ± 12.1/889.8 ± 66.2	84.3 ± 0.4
H	4.48 ± 0.16	77.7 ± 0.7	(1.06 ± 0.01) × 10^6^	105.9 ± 5.1/476.4 ± 23.6	57.8 ± 1.2

## Data Availability

Data are contained within the article.

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
