# Peer review of "Foaming of Bio-Based PLA/PBS/PBAT Ternary Blends with Added Nanohydroxyapatite Using Supercritical CO_2_: Effect of Operating Strategies on Cell Structure"

_molecules, 2025, doi:10.3390/molecules30092056_

Round 1
Reviewer 1 Report
Comments and Suggestions for Authors
In this manuscript, the authors reported the foaming of bio-based PLA/PBS/PBAT ternary blends with added nanohydroxyapatite using supercritical CO2: Effect of operating strategies on cell structure. Overall, the research topic is kind of interesting. I would therefore support publication, but before that, certain technical issues need to be addressed (see below for more detailed comments).
- The novelty of this study is unclear and should be clearly stated and justified.
- Why is there no correlation drawn between cell morphology and mechanical properties?
- The author must provide the thermal properties, particularly the melting behavior of the prepared samples.
- If possible, provide the antimicrobial activity and cell viability results of the fabricated samples.
- Are these kinds of foam materials useful for industry, or are they only for academic study?
Author Response
Reply to the comments of Reviewer #1
Comments and Suggestions for Authors
In this manuscript, the authors reported the foaming of bio-based PLA/PBS/PBAT ternary blends with added nanohydroxyapatite using supercritical CO2: Effect of operating strategies on cell structure. Overall, the research topic is kind of interesting. I would therefore support publication, but before that, certain technical issues need to be addressed (see below for more detailed comments).
Reply: We appreciate very much the kind comment from Reviewer 1.
1. The novelty of this study is unclear and should be clearly stated and justified.
Reply: We appreciate very much the kind comment of Reviewer 1. In the last paragraph of the Introduction section, we have stated that to the best of our knowledge, SCCO2 foaming of PLA/PBS/PBAT ternary polymer blends with added nHA has not been shown in the literature. In our revised manuscript, we have added the statement at the end of the Introduction section (page 3): The innovation of this study is the foaming of a ternary polymer blend with the addition of nHA, leading to the formation of bimodal cell structure by employing a gradual pressure reduction process.
2. Why is there no correlation drawn between cell morphology and mechanical properties?
Reply: We appreciate very much the kind comment of Reviewer 1. We measured the compressive strength of the foam products, as shown in Figure 11, and stated that the foam products with larger macrocell sizes exhibited better toughness. On the other hand, the foam products with smaller macrocell sizes showed better stiffness. The qualitative correlation between cell size/morphology and mechanical property has been demonstrated.
3. The author must provide the thermal properties, particularly the melting behavior of the prepared samples.
Reply: We appreciate very much the kind comment of Reviewer 1. In our previous study: Bio-based PLA/PBS/PBAT ternary blends added with nanohydroxyapatite: a thermal, physical and mechanical study, Polymers, 2023, 15, No. 4585 (Reference [33] in our manuscript), we have provided the detail thermal properties, including the melting behavior of the samples used in this study (PLA/PBS/PBAT, 80/15/5 wt%, added with 5 wt% nHA). We have added this statement in the Introduction part (page 2) of our revised manuscript.
4. If possible, provide the antimicrobial activity and cell viability results of the fabricated samples.
Reply: We appreciate very much the kind comment of Reviewer 1. The main effort of this study is to investigate the formation of bimodal cell structures using an adequately designed foaming temperature and pressure strategy. The antimicrobial activity and cell viability study will be our future research goals.
5. Are these kinds of foam materials useful for industry, or are they only for academic study?
Reply: We appreciate very much the kind comment of Reviewer 1. The foam materials in this study were produced using small-scale laboratory equipment, as described in Section 3.3. of our manuscript. Currently, these foam materials are suitable for academic study.
Reviewer 2 Report
Comments and Suggestions for Authors
The study presents the foaming behavior of a composite polymer blend under various foaming pressure and temperature conditions. The manuscript contains interesting foaming approaches and vast characterization results. The explanation of the observations is reasonable.
Some issues that require attention are:
- There is no comment/characterization result on the Hydroxyapatite distribution in the polymer matrix. How large were the agglomerated particles?
- There is no comment/characterization result on the polymer blend compatibility. Was the polymer matrix composition uniform? Sometimes, the bimodal distribution of pores arises from the nonuniformity of the polymer matrix.
- Abbreviations, e.g. PLA/PBS/PBAT, should be explained in abstract.
- How was the temperature remained constant during the pressure drop ? Please add experimental details, or mention any uncertainty, because due to the Joule-Thomson effect a remarkable temperature reduction is observed, which sometimes results even in solid CO2 production.
Author Response
Reply to the comments of Reviewer #2
Comments and Suggestions for Authors
The study presents the foaming behavior of a composite polymer blend under various foaming pressure and temperature conditions. The manuscript contains interesting foaming approaches and vast characterization results. The explanation of the observations is reasonable.
Reply: We appreciate very much the kind comment from Reviewer 2.
Some issues that require attention are
1. There is no comment/characterization result on the Hydroxyapatite distribution in the polymer matrix. How large were the agglomerated particles?
Reply: We appreciate very much the kind comment of Reviewer 2. The results of characterization on the Hydroxyapatite distribution in the polymer matrix have been described in our previous study (Reference [33] in our manuscript): Bio-based PLA/PBS/PBAT ternary blends added with nanohydroxyapatite: a thermal, physical and mechanical study, Polymers, 2023, 15, No. 4585. Our previous study showed that 5 wt% nHA was uniformly distributed in the polymer blends, as shown in the TEM measurement image. We added the following statement in the Introduction part (page 2) of our revised manuscript: Our previous study [33] stated that the 5 wt% added nHA was uniformly distributed in the polymer blend from the TEM measurement image.
2. There is no comment/characterization result on the polymer blend compatibility. Was the polymer matrix composition uniform? Sometimes, the bimodal distribution of pores arises from the nonuniformity of the polymer matrix.
Reply: We appreciate very much the kind comment of Reviewer 2. In our previous study (Polymers, 2023, 15, No. 4585, Reference [33] of this manuscript), it was indicated that by adding 5 wt% nHA, the uniform distribution of the polymer phase, between the continuous PLA matrix and the dispersed phase of PBS/PBAT was improved. Cell nucleation may initiate at the interface between polymer phases. However, as shown in Figures 1 and 2 of our manuscript, foaming strategies A, B, and C produced unimodal cell size distribution. This indicated that PBS/PBAT was uniformly dispersed in the continuous PLA phase. The bimodal cell structures in Figures 4 and 6 of our manuscript only appeared after changing the operating conditions through a stepwise pressure reduction process.
3. Abbreviations, e.g. PLA/PBS/PBAT, should be explained in abstract.
Reply: We appreciate very much the kind comment of Reviewer 2. We have added the full names of PLA, PBS, and PBAT to our revised manuscript's abstract (page 1).
4. How was the temperature remained constant during the pressure drop? Please add experimental details, or mention any uncertainty, because due to the Joule-Thomson effect a remarkable temperature reduction is observed, which sometimes results even in solid CO2.
Reply: We appreciate very much the kind comment of Reviewer 2. A self-made heater controlled the temperature. Since the foaming apparatus was small, it was relatively easy to control the constant temperature during the pressure drop by regulating the temperature controller. The uncertainty of the temperature was ± 2 °C. We have added this statement in our revised manuscript (page 13).
Round 2
Reviewer 1 Report
Comments and Suggestions for Authors
Authors have addressed all the major concerns, so I would recommend it for publication.